# Photophysical Properties of Multilayer Graphene–Quantum Dots Hybrid Structures

**DOI:** 10.3390/nano10040714

**Published:** 2020-04-09

**Authors:** Ivan Reznik, Andrey Zlatov, Mikhail Baranov, Roman Zakoldaev, Andrey Veniaminov, Stanislav Moshkalev, Anna Orlova

**Affiliations:** 1School of Photonics, ITMO University, 197101 St. Petersburg, Russia; ivanreznik1993@mail.ru (I.R.); zlatov.andrei@gmail.com (A.Z.); mbaranov@mail.ru (M.B.); zakoldaev@gmail.com (R.Z.); avveniaminov@itmo.ru (A.V.); 2Center for Semiconductor Components, State University of Campinas (Unicamp), Campinas 13083-970, Brazil; stanisla@unicamp.br

**Keywords:** multilayered graphene, CdSe–ZnS quantum dots, hybrid structures, photophysical properties, photoelectrical properties, photoactivation

## Abstract

Photoelectrical and photoluminescent properties of multilayer graphene (MLG)–quantum dots (QD) hybrid structures have been studied. It has been shown that the average rate of transfer from QDs to the MLG can be estimated via photoinduced processes on the QDs’ surfaces. A monolayer of CdSe QDs can double the photoresponse amplitude of multilayer graphene, without influencing its characteristic photoresponse time. It has been found that efficient charge or energy transfer from QDs to MLG with a rate higher than 3 × 10^8^ s^−1^ strongly inhibits photoinduced processes on the QD surfaces and provides photostability for QD-based structures.

## 1. Introduction

Quantum semiconductor nanocrystals (NCs) (i.e., crystals confined to a few nanometers in one, two, or three dimensions) are among the most popular and fast-evolving species in current research [1]. Starting half a century ago with the first epitaxial quantum wells in a silicon substrate, investigation and synthesis of nanostructured materials has led to the formation of a vast variety of different nanostructured systems [2]. NCs reveal their superiority over bulk materials through their unique mechanical, optical, electrical, magnetic, acoustic, and biological properties [1,2,3,4,5,6]. Despite the increasing number of nanostructured materials that are newly synthesized each year, their practical application has, however, been very limited to date [7,8,9]. A lack of models describing key excitonic relaxation paths in NCs is one of the reasons for their limited application, in addition to the low stability and efficacy of NC-based devices. Some of the vivid examples of such materials are 0D semiconductor nanocrystals, referred to as quantum dots (QDs) [10,11]. Today, QD utilization in industry is restricted to the creation of high-performance LEDs and lasers [12,13]; however, QDs appear to be superior in many practical applications, such as environmental monitoring, sensorics, solar energy utilization, biology, and medicine [14,15,16]. 

The peculiar features of semiconductor QDs are the dimensional dependence of their optical properties and their high photoluminescence (PL) quantum yield (QY). Additionally, the QD surfaces can be easily functionalized, allowing the design of systems in which QDs are utilized as efficient donors of energy or charge to the nearest acceptor [17,18,19]. At the same time, low carrier mobility in QDs is usually compensated for by either the geometry of the QD-based photovoltaic systems [20] or by combining QDs with high carrier mobility materials, such as MoS_2_ and graphene [21,22]. Hybrid systems based on graphene and semiconductor QDs are the most popular solutions for achieving high efficiency in the conversion of absorbed energy into useful signals for the final photoelectric device [23,24,25,26].

The instability of the hybrid graphene–QDs structure parameters under prolonged and powerful external radiation is one of the main problems for photovoltaic devices based on QDs [27]. Photoinduced changes in the number and efficiency of trap states on the QDs’ surfaces can significantly change their optical and electrical properties [28]. All photoinduced processes on the QDs’ surfaces can be split into two groups, resulting in either photoactivation or photodegradation of QDs [29]. Photoactivation is accompanied by an increase in QD excitonic luminescence because of passivation of dangling chemical bonds on the QD surfaces, and by a decrease in the number of trap states. Photoactivation can start immediately after switching on the external irradiation. It should be pointed out that the efficiency of QD photoactivation depends on (i) the QD surfaces properties (i.e., number of trap states), (ii) QD type (i.e., core or core–shell QDs), and (iii) environment (i.e., oxygen and water concentrations) [30,31,32]. QD photodegradation is associated with the QD surfaces oxidation, which leads to shifts of QD excitonic absorption and emission bands of higher energy due to destruction of the QD outer layers and an increase in the number of trap states. Despite the influence of external radiation on the efficiency of QD photoactivation and photodegradation processes having been studied in several works [33,34], systematic studies of external irradiation effects on the physical properties of QD-based structures are still needed.

In this paper, we describe the influence of external irradiation on photoluminescence and photoelectric properties of multilayer graphene–QD (MLG–QD) hybrid structures deposited onto titanium contacts. We demonstrate for the first time that the average transfer rate from QDs to MLG can be estimated via photoinduced processes on the QDs’ surfaces. We show that the CdSe QD monolayer can double the photoresponse amplitude of MLG, while its characteristic photoresponse time is not influenced. We demonstrate that efficient charge or energy transfer from QDs to MLG with a rate higher than 3 × 10^8^ s^−1^ should strongly inhibit photoinduced processes on the QDs’ surfaces and provide photostability for QD-based structures.

## 2. Materials and Methods

### 2.1. Materials

*N*-methyl-2-pyrrolidone (NMP), chloroform, ethanol, toluene, and hexane were purchased from Sigma-Aldrich (St. Louis, MO, USA) and used without further purification. First-type colloidal CdSe–ZnS QDs synthesized by the hot injection organometallic method described in [35] were used for the formation of hybrid structures, with multilayer graphene (MLG) supplied by the Nacional de Grafite company (São Paulo, Brazil). The QDs had a 5.5 nm average core diameter. MLG used in this study usually contained nanobelts with thicknesses varying from a few nanometers to tens of nanometers, with a width of 5–10 µm and length of 10–50 µm [36].

### 2.2. Ligand Exchange Procedure

QDs originally passivated with trioctylphosphine oxide (TOPO) and oleic acid (OA) molecules were synthesized in accordance with the protocol described in [35], which resulted in an indefinite ratio of OA to TOPO on their surface. Such a mixture of ligand molecules on the QD surfaces tends to obstruct uniform formation of Langmuir–Blodgett (L–B) films [37]. Therefore, after the synthesis, we replaced TOPO molecules on the QD surfaces with a more uniform layer of OA molecules using the following technique. Firstly, the initial mixture of stabilizer molecules on the QD surfaces was removed by washing and precipitating a QD solution in a 1:1:1 mixture of methanol, acetone, and chloroform by centrifugation at 5000 rpm for 5 min. In the next step, the QDs were dissolved in a concentrated solution of OA in chloroform and left in the dark for 24 h. Finally, the QD solution was precipitated using a methanol/acetone/chloroform mixture and redissolved in a fresh portion of chloroform. 

### 2.3. Formation of MLG–QD Hybrid Structures Layered on Slides

Two types of glass slides were used as substrates for the preparation of MLG–QD hybrid structures. Firstly, all the slides were kept in a chromic acid solution for 24 h. Then, the slides were both washed and kept in a closed bottle with distilled water. Next, a part of the slides was coated by a 300 nm thick titanium layer using magnetron sputtering of a titanium target in a Kurt Lesker PVD 75 deposition unit (Kurt J. Lesker, Jefferson Hills, PA, USA). Finally, pairs of planar contacts were formed by creation of 100 μm gaps in the titanium layer using laser ablation (Mini-Marker 2, St. Petersburg, Russia). MLG–QD hybrid structures were formed on top of both types of slides (covered and not covered by Ti) using the L–B deposition method [38]. Samples with and without titanium planar contacts were used to study photoelectrical and photoluminescent properties of structures, respectively. Continuous QD layers and MLG layers deposited separately on the glass slides were used as reference samples. Table 1 presents the characteristics of the studied samples. 

### 2.4. Characterization of the Structures

Analysis of the PL kinetics of QDs was done using a fluorescence microscope equipped with a MicroTime100 time-correlated single-photon counting spectrometer (Pico Quant, Berlin, Germany) with a 409 nm laser. Analysis of the absorbance and fluorescence of colloidal QDs was carried out using a UV probe 3600 spectrophotometer (Shimadzu, Kyoto, Japan) and a Cary Eclipse luminescence spectrophotometer (Varian, Palo Alto, California, USA). The morphology of QD monolayers was investigated using an LSM-710 confocal microscope (Carl Zeiss AG, Oberkochen, Germany), Merlin scanning electron microscope (Carl Zeiss AG, Oberkochen, Germany), and a Solver PRO-M atomic force microscope (NT-MDT, Moscow, Russia). The photoelectric properties of MLG–QD hybrid structures were studied using a 405 nm/5 mW laser and a Keithley 2636B picoampermeter (Keithley Instruments, Cleveland, OH, USA), synchronized with a microcontroller. The voltage applied to pairs of electrodes in all experiments was equal to 5 volts.

The photoinduced change in the state of the QD surface was induced by controlled irradiation with the 405 nm/5 mW CW laser. The incident laser radiation power was monitored and controlled using a Thorlabs S130C power meter and a polarizing filter. The power of the microscope’s laser excitation during measurements was several orders of magnitude lower than the laser radiation power during photo-irradiation of QDs.

The diameter of the QD core was calculated using Peng’s empirical formula [39]. Taking into account the correction for the ZnS shell, the diameter of quantum dots with a luminescence maximum wavelength of 635 nm was estimated to be 5.5 nm.

The PL decay was approximated by biexponential function according to Equation (1):(1)$$y=y_{0}+A_{1}\cdot \exp\left(-\frac{\left(x-x_{0}\right)}{t_{1}}\right)+A_{2}\cdot \exp\left(-\frac{\left(x-x_{0}\right)}{t_{2}}\right)$$
where *A_i_* is the amplitude of the *i*th (1st or 2nd) decay component and *τ_i_* is the characteristic decay time of the *i*th component.

Amplitude-weighted averaging of the QDs’ PL decay time was calculated according to Equation (2):(2)$$\langle \tau \rangle =\frac{A_{1}\tau_{1}^{2}+A_{2}\tau_{2}^{2}}{A_{1}\tau_{1}+A_{2}\tau_{2}}$$

The QDs’ PL quantum yield was estimated using rhodamine 6G dissolved in ethanol (95% quantum yield of PL) as a reference, according to Equation (3):(3)$$\frac{\varphi_{smpl}}{\varphi_{ref}}=\frac{I_{smpl}\cdot n_{smpl}^{2}\cdot D_{ref}}{I_{ref}\cdot n_{ref}^{2}\cdot D_{smpl}}$$
where *φ_smpl_* and *φ_ref_* are the quantum yields for QDs and rhodamine 6G luminescence, respectively; *I_smpl_* and *I_ref_* are the intensities at the maxima of the luminescence band for QDs and rhodamine 6G, respectively; *D_smpl_* and *D_ref_* are optical densities at the luminescence excitation wavelength for QDs and rhodamine 6G, respectively; *n_smpl_* and *n_ref_* are the refractive indices of the solvents toluene and ethanol, respectively [40].

## 3. Results and Discussion

The optical properties of QDs stabilized with OA molecules and PL properties, and the morphology of QD monolayers on dielectric slides and in MLG–QD hybrid structures were studied using FTIR (Appendix A), steady-state UV and PL spectroscopy (Appendix A, SEM (Appendix A), Atomic Force Microscopy (AFM) (Appendix A), and Laser Scanning Microscopy (LSM) (Appendix A) techniques (see Appendix A for details).

### 3.1. PL Kinetics of CdSe QDs in MLG–QD Structures

Figure 1 presents PL decay curves for QDs in monolayers on a dielectric glass slide and in MLG–QD structures on Ti contacts. The notable decrease (by an order of magnitude) in the PL intensity of QDs in the MLG–QD hybrid structures (Appendix A) was accompanied by shortening of the QD exciton PL decay time, as clearly seen in Figure 1.

The PL decay of QDs on a dielectric slide and in MLG–QD structures presented in Figure 1 was fitted with the biexponential functions (Equation (1)); the QD subensembles with shorter and longer relaxation times, τ_1_ and τ_2_, are further referred to as fraction 1 and fraction 2, respectively. 

The reasons for the biexponential PL decay of CdSe QDs at room temperature are still unclear, and debate regarding the underlying mechanisms continues to date. In particular, there have been attempts to explain the complex decay via PL blinking by extra charges in QDs or trap states on the QD surfaces, which can deactivate excitons [41]. At the moment, it is not possible to determine which of these two mechanisms dominates in our hybrid structures. 

It is well known that the PL decay time and the PL QY of 1st and 2nd QD PL fractions can be described as follows:(4)$$\tau_{i}=\frac{1}{k_{ri}+k_{nri}}$$
(5)$$\varphi_{i}=k_{ki}\cdot \tau_{i}$$
where *k_ri_* and *k_nri_* are radiative and nonradiative rates of exciton deactivation, respectively; *τ_i_* and *φ_i_* are the PL decay time and QY of the 1st and 2nd QD fractions, respectively. 

We analyze two QD fractions that are characterized by different PL decay times separately in this paper. We suppose that these QD fractions have the same radiative rate (i.e., *k_ri_* = *k_r_* = 4∙10^7^ s^−1^), according to [42]. Therefore, the difference in PL decay time is caused by different nonradiative rates (*k_nri_*) only. In our QD monolayer samples, both QD PL fractions with τ_1_ ~2 ns and τ_2_ ~7 ns give approximately the same contribution to the PL signal. A schematic representation of various radiative and nonradiative relaxation pathways of the excited state in QDs for a MLG–QD hybrid structure system is shown in Figure 2.

Analysis of the characteristic PL lifetimes and amplitudes of the QD PL components has shown that QDs in both PL fractions are strongly quenched in MLG–QD structures due to efficient charge or energy transfer from QDs to MLG [43,44,45]. Due to the fact that the characteristic amplitudes correlate with the quantity of luminescent QDs and the observed decrease in PL amplitude by almost two orders of magnitude (from ~1300 to ~50 counts, according to inset table in Figure 1), we can suppose that we detected the luminesce signal only from approximately 3–4% in the whole QD ensemble. This clearly confirms the interaction of QDs and MLG in the structures because of the charge or energy transfer from QD to MLG. It should also be pointed out that the PL quenching efficiency of QDs should be estimated without omission of totally quenched QDs. Therefore, we propose here to compare the PL intensities of QDs in the structures with MLG and QD monolayers on dielectric slides for estimation of the PL quenching efficiency of QDs, as follows:(6)$$Q^{G}=1-\frac{I}{I_{0}}=1-\frac{\sum_{i=1}^{k}A^{i}\cdot \frac{\tau^{i}}{\tau_{r}^{i}}}{\sum_{i=1}^{k}A_{0}{}^{i}\cdot \frac{\tau_{0}{}^{i}}{\tau_{r_{0}}^{i}}}$$
where *I* and *I*_0_ are QD PL intensities after and before interaction with MLG; *A^i^* is the amplitude of the *i*th QD fraction; *τ^i^* is the PL decay time of the *i*th QD fraction; *τ_r_* is the radiative time of CdSe QDs, which is equal to 25 ns [32].

The correct estimation of PL quenching of QDs in hybrid structures, where QDs are used as energy or charge donors with Equation (6), is valid under the following conditions only: (i) the QD numbers in the region of interest (region of interest) in the structure and reference samples have to be equal; (ii) all experimental parameters, such as the laser excitation repetition rate, should be the same for PL data acquisition from the structure and the reference samples. 

We can estimate the rate of interaction of a QD fraction (*k_i_^G^*) with MLG in the structures, as follows:(7)$$k_{i}^{G}=\frac{1}{\tau_{i}^{G}}-\left(k_{r}+k_{nr_{i}}\right)$$
where *τ_i_^G^* is the PL decay time of the *i*th QD fraction in hybrid structures.

QD parameters presented in Table 2 clearly demonstrate that both QD luminescent fractions are characterized by a high charge and energy transfer efficiency towards MLG, despite the strong difference in the rates (*k_i_^G^*) of their interactions with MLG (*Q*_i_^G^ referring to Table 2, showing 98% and 97.5% for 1st and 2nd QD fractions, respectively). It should be noted that using Equation (7), we can estimate the minimal rate of QD interaction with MLG that totally quenches the PL of QDs in the hybrid structures, which is no less than 0.4 × 10^8^ s^−1^.

### 3.2. Photoelectric Properties of MLG–QD Hybrid Structures

Figure 3 shows how the photoresponses of MLG–QD hybrid structures and MLG are changed under the sample irradiation with a 405 nm LED during periods of ~60 s uniform irradiation. It is clearly shown that QDs improve the photoresponse of MLG strongly because of their interactions in hybrid structures. It is unlikely that energy transfer from QDs to monolayer graphene affects the conductivity of QD–graphene hybrid structures, because the defection-free graphene does not demonstrate any significant photoresponse [45]. At the same time, as clearly seen in Figure 3a, our MLG samples also increase their conductivity under external irradiation. This means that the irradiation of the MLG samples leads to a charge injection. The high density of the trap states on the surface of the MLG caused by sonication during their formation [46] may be the reason for the photoresponse of MLG. The photoresponse from MLG indicates that energy transfer from the QD monolayer to MLG in our hybrid structures can also change the conductivity of the MLG layer. Until now, there has been no reason to believe that an energy transfer channel can lead to photoresponse in the MLG–QD hybrid structure. Additionally, there is no direct experimental evidence to distinguish between the effects of energy and charge transfers in the photoresponse of MLG–QD structures.

Figure 3b shows the enlarged front part of the photoresponse from MLG–QD hybrid structures and MLG. Stage I represents the dark conductivity of the samples. Stages II and III of the sample photoresponse can be fitted by the biexponential function:(8)$$y=y_{0}+A_{1}\cdot \exp\left(\frac{x}{t_{1}}\right)+A_{2}\cdot \exp\left(\frac{x}{t_{2}}\right)$$
where *A*_i_ is the amplitude and *t*_i_ is the characteristic time for stage I. The fitting parameters of the photoresponses for MLG and hybrid structures are presented in Table 3.

As can be seen from Table 3, MLG and hybrid structures demonstrate very similar characteristic times for stages II and III of about 0.3 and 5.25 s, respectively. These times are in good agreement with the previously published works on the photoelectric properties of hybrid structures based on MLG and QDs [47]. Stage II is the rapid growth phase of the photoresponse and is typically attributed to the direct transport of photogenerated charge carriers to contacts from quantum dots. Stage III is traditionally discussed [48] as the slow increase phase of the photoresponse caused by the slow transport rate of charge carriers in the QD layer lying within the substrate regions not covered by graphene. It should be pointed out that in our samples, the characteristic times for stages II and III in MLG and MLG–QDs are almost equal. At the same time, the photoresponse amplitudes for hybrid structures are doubled for both stages in comparison to MLG. This confirms the existence of effective charge or energy carrier transfer from QDs to MLG. This also clearly demonstrates that the characteristic times for stages II and III (see Figure 3b) are fully determined by the electric properties of MLG and the architecture of the structures. This result refutes the traditional explanation of the differences in photoresponse rates at stages II and III in hybrid structures based on MLG and QDs [49].

### 3.3. Photoactivation of MLG–QD Hybrid Structures

The photoinduced change of physicochemical properties of the QD surface—so-called photoactivation—is a well-known phenomenon in colloidal QDs that can significantly change the QY of their PL due to change of the nonradiative exciton relaxation rate in the QDs [50]. It is obvious that changes in the nonradiative rate in QDs can alter the functionality of QD-based hybrid structures acting as energy or charge carrier donors. Therefore, we studied the impact of QD photoactivation on photoelectrical properties of MLG–QD hybrid structures. Figure 4 shows the dependence of the photophysical and photoelectric properties of QDs deposited on a dielectric slide and included in the MLG–QD hybrid structure before and after QD photoactivation.

The comparison of PL properties of the QD monolayer on a glass slide before and after photoactivation with a 72 J/cm^2^ dose showed up to 1.5-fold growth of both the average PL decay time calculated using Equation (2) and the PL intensity. This means that irradiation of QD monolayer samples on dielectric slides by 72 J/cm^2^ at 405 nm light leads to a decrease in the nonradiative rate in QDs from 6.9 × 10^7^ s^−1^ to 2.7 × 10^7^ s^−1^, according to Equations (4) and (5). At the same time, as clearly seen from Figure 4a, there are no changes in the PL decay time or PL intensity of QDs in the MLG–QD hybrid structures. The efficiency of QD photoactivation processes (*Q^PA^*) calculated with Equations (S1) and (S2) clearly decreases by an order of magnitude for QDs located on MLG, in comparison with QDs located on dielectric substrates (i.e., *Q^PA^* of ~100% for QDs on dialectic substrates vs. *Q^PA^* of ~10% for MLG–QDs). This means that the rate of charge or energy transfer from QDs to MLG is much larger than the rate of nonradiative exciton relaxation processes in which trap states are involved. This allows us to estimate the minimal average charge or energy transfer rate from QDs to MLG in our hybrid structures as 〈*k^G^*〉 ≥ 1.0 × 10^8^ s^−1^(according to Table 2). This value demonstrates an excellent agreement with the k^G^ value estimated for the 1st QD luminescent fraction using PL quenching of QDs in the MLG–QD hybrid structures. This also implies that the *k^G^* value for both QD luminescent fractions is at least 1.0 × 10^8^ s^−1^ (see Table 2).

## 4. Conclusions

Our results clearly demonstrate that efficient charge or energy transfer from the CdSe–ZnS QD monolayer to multilayer graphene with a rate higher than 1.0 × 10^8^ s^−1^ is possible. We show that efficient interaction of QDs with MLG in our MLG–QDs hybrid structures enhances the photoresponse of these structures by a factor of up to 1.5 with respect to MLG. The analysis of the electric properties of the hybrid structures shows that the characteristic photoresponse time of the hybrid structures depends only on the electric properties of MLG and on the architecture of the hybrid structures. We demonstrate for the first time that QD photoactivation in hybrid structures can be an efficient tool for the estimation of the interaction rate of QDs–MLG.

## Figures and Tables

**Figure 1 nanomaterials-10-00714-f001:**
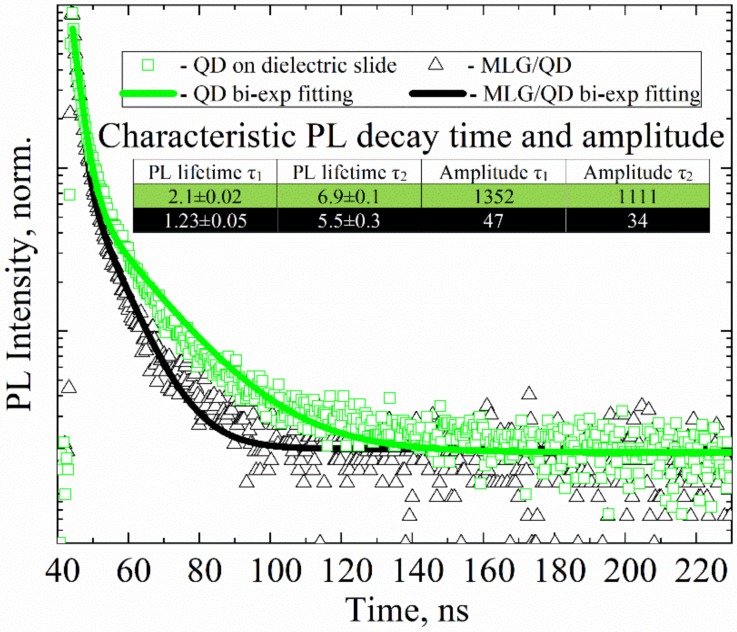
Photoluminescence (PL) decay curves of 5.5 nm CdSe–ZnS quantum dots (QDs) on a dielectric slide (green rectangles) and in the multilayer graphene (MLG)–QD structures (black triangles). A 405 nm pulse laser was used for PL excitation of QDs. Solid lines (green and black) are biexponential fits of the decay. Inset: table showing the fitting parameters of PL decay curves.

**Figure 2 nanomaterials-10-00714-f002:**
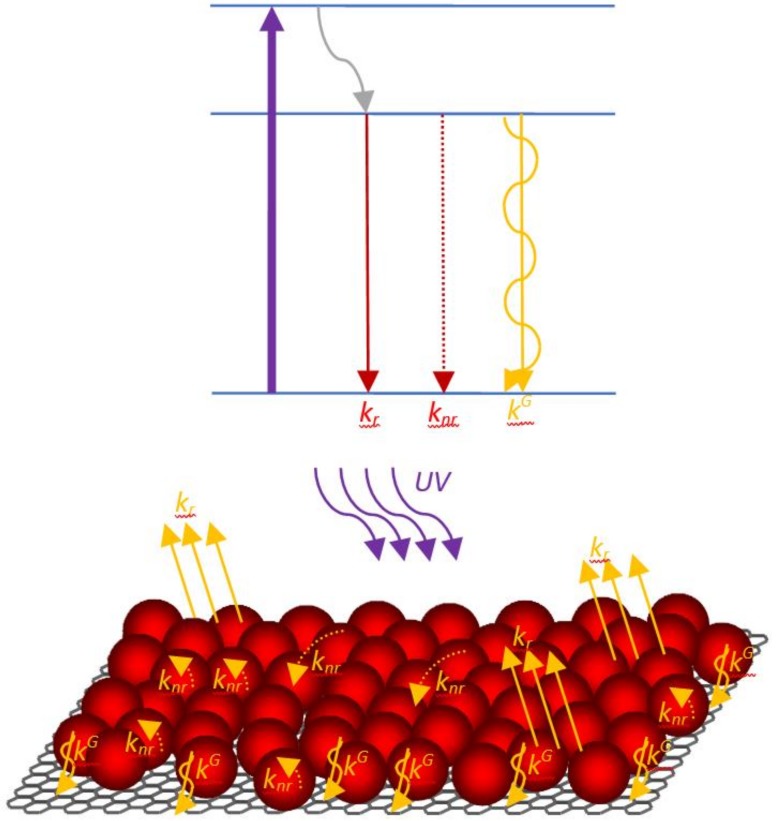
Schematic representation of radiative and nonradiative pathways of the excited state in QDs.

**Figure 3 nanomaterials-10-00714-f003:**
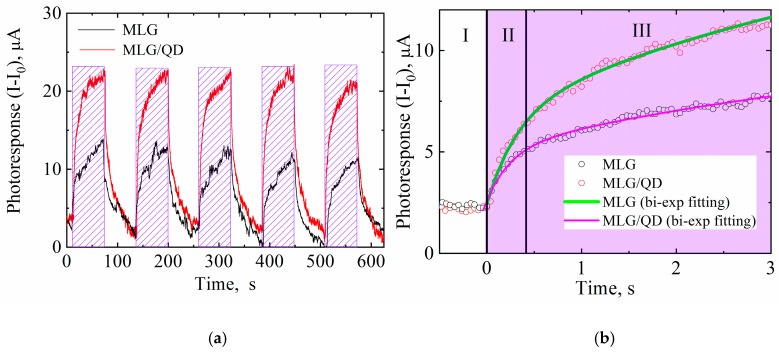
(**a**) The photoresponse of MLG (black line) and MLG–QD hybrid structures (red line) with irradiation with a 405 nm laser (dashed regions). (**b**) Enlarged parts of the curves (black and red circles) with biexponential fitting (green and purple line).

**Figure 4 nanomaterials-10-00714-f004:**
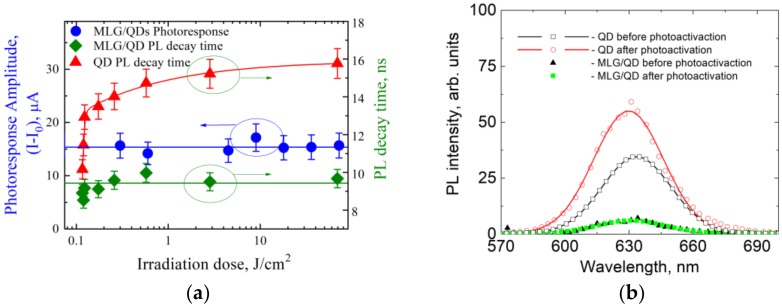
Photoactivation of QD monolayer and MLG–QD hybrid structures. (**a**) Irradiation dose dependence of the photoresponse amplitude of MLG–QD structures (blue spots) and of the average PL decay time of QDs layered on dielectric slides (red triangles) and in the hybrid structures (green rectangles). Solid lines are guidelines for the eye. (**b**) Photoluminescence (PL) spectra of QDs before photoactivation (black rectangles and triangles) and after exposure to 72 J/cm^2^ of UV light (red circles, green rectangles) on dielectric slides (black rectangles, red circles) and in the hybrid structures (black triangles, green rectangles).

**Table 1 nanomaterials-10-00714-t001:** Characteristics of the studied samples. MLG, multilayered graphene; QDs, quantum dots.

Samples	Characteristics
MLG	Multilayered graphene nanobelts with 30–40 nm thickness deposited on a glass slide with titanium contacts
QDs	5.5 nm core–shell CdSe–ZnS QD monolayer deposited on a glass slide from toluene solution
MLG–QDs	Hybrid structures with 5.5 nm core–shell CdSe–ZnS QD monolayer deposited on the MLG on a glass slide with titanium contacts

**Table 2 nanomaterials-10-00714-t002:** The photophysical properties of CdSe–ZnS QD monolayers on glass slides and in hybrid structures based on MLG.

Samples	Parameters	Units	1st QD fraction	2nd QD fraction	Formula
CdSe–ZnS QDs monolayer	*k_r_*	s^−1^	0.4 × 10^8^		-
	*φ_i_*	%	8.4	27.7	(5)
	*τ_i_^QD^*	ns	2.1 ± 0.1	6.9 ± 0.1	-
	*A_i_*	Counts	1350 ± 50	1100 ± 50	-
	*k_nr_*	s^−1^	4.3 × 10^8^	1.0 × 10^8^	(4)
Hybrid structures	*τ_i_^G^*	ns	1.2 ± 0.1	5.5 ± 0.3	-
	*A_i_*	Counts	47 ± 2	34 ± 2	-
	*k_i_^G^*	s^−1^	(3.4 ± 0.2) × 10^8^	(0.40 ± 0.05) × 10^8^	(7)
	*Q_i_* ^G^	%	98 ± 2	97.5 ± 2	(6)
	*Q* ^G^	%	98 ± 2		(6)

**Table 3 nanomaterials-10-00714-t003:** Fitting parameters of the photoresponses of MLG and MLG–QD hybrid structures ^1^.

Samples	Stage II		Stage III	
	*A*_II,_ μA	*t*_II,_ s	*A*_III,_ μA	*t*_III,_ s
MLG	2.5 ± 0.15	0.25 ± 0.03	5.9 ± 0.1	5.1 ± 0.3
MLG–QD Hybrid structures	4.4 ± 0.15	0.31 ± 0.02	11.1 ± 0.1	5.3 ± 0.2

^1^ The fitting parameters *A*_i_ and *t*_i_ are obtained with Equation (8).

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
