# Peer review of "Photophysical Properties of Multilayer Graphene–Quantum Dots Hybrid Structures"

_nanomaterials, 2020, doi:10.3390/nano10040714_

Round 1
Reviewer 1 Report
The manuscript describes a hybrid structure based on graphene and classical semiconductor quantum dots (QDs). Such structures have a high potential in photovoltaic, energy storage and sensor systems. The authors have suggested a new approach to estimate the rate of QDs interaction with graphene in the hybrid structures based on the phenomenon of QD photoactivation. The authors demonstrated that not only the relaxation time, but also the amplitudes of photoluminescence decay should be taken into account for correct estimation of phosphor quenching, when the major part of the phosphor is totally quenched. The manuscript presents a novel and interesting results, corresponds to the journal profile, and it is of interest to numerous specialists in the field of nanostructured matter research.
At the same time, some authors’ statements should be clarified before publications:
1) The authors should explicitly describe the conditions under which QD quenching estimation using Eq. (6) is valid.
2) The authors discuss energy/charge rate assuming that not only charge transfer, but also energy transfer from QDs to graphene can lead to increase of charge carrier concentration in graphene within hybrid structures. I suggest that clarification of this assumption is needed, since most other authors consider only charge transfer effect on conductivity.
3) Authors should more clearly distinguish between explanations for photoresponse rates in stages II and III in Fig. 3 (b), proposed by them in the manuscript, and explanations generally accepted by other researchers.
4) In addition some more clarifications should be given in the manuscript. In particular, CdSe/ZnS QDs and CdSe/ZnS QD monolayer should be shown in Table 1 and Table 2 instead of QDs and CdSe/ZnS monolayer, respectively.
Therefore, from my point of view this paper can be published in journal Nanomaterials after minor revision.
Author Response
Comment 1
“The authors should explicitly describe the conditions under which QD quenching estimation using Eq. (6) is valid.”
We have added some sentences on Page 6 in the manuscript according to the comment.
"The correct estimation of PL quenching of QDs in hybrid structures where QDs are used as energy or charge donors with Eq. (6) is valid under the following conditions only: (i) the QD numbers in the ROI, i.e. region of interest, in a structure and in reference samples have to be equal; (ii) all experimental parameters such as laser excitation repetition rate, etc. should be the same for PL data acquisition from the structure and the reference samples."
Comment 2
“The authors discuss energy/charge rate assuming that not only charge transfer, but also energy transfer from QDs to graphene can lead to increase of charge carrier concentration in graphene within hybrid structures. I suggest that clarification of this assumption is needed since most other authors consider only the charge transfer effect on conductivity.”
We have added several sentences on Page 7 in the manuscript according to the comment.
"It is unlikely that energy transfer from QDs to monolayer graphene affects the conductivity of QD-graphene hybrid structures because the defect free graphene doesn't demonstrate any significant photoresponse [Xia, F., Mueller, T., Lin, Y. M., Valdes-Garcia, A., &Avouris, P. (2009). Ultrafast graphene photodetector. Nature nanotechnology, 4(12), 839.]. At the same time as clearly seen in Fig.3a, our MLG samples also increase their conductivity under external irradiation. It means that the irradiation of the MLG samples leads to charge injection. A high density of trap states on the surface of the MLG caused by sonication during their formation [46] may be the reason for the photoresponse of MLG. Photoresponse from MLG indicates that energy transfer from QD monolayer to MLG in our hybrid structures can also change the conductivity of the MLG layer. Up to now, there are no reasons to believe that in MLG/QD hybrid structure no energy transfer channel exists. As well as there is no direct experimental evidence to distinguish between the effects of energy and charge transfer in the photoresponse of MLG/QDs structures."
Comment 3
“Authors should more clearly distinguish between explanations for photoresponse rates in stages II and III in Fig. 3 (b), proposed by them in the manuscript, and explanations generally accepted by other researchers”
We have rewritten several sentences on page 8 in the manuscript according to the reviewer's comment.
"Stage II is the rapid-growth phase of the photoresponse and it is typically attributed to the direct transport of photo-generated charge carriers to contacts from quantum dots. The stage III is traditionally discussed [48] as the slow-rise phase of the photoresponse caused by slow rate transport of charge carriers in the QD layer lying within the substrate regions not covered by graphene."
Comment 4
"In addition, some more clarifications should be given in the manuscript. In particular, CdSe/ZnS QDs and CdSe/ZnS QD monolayer should be shown in Table 1 and Table 2 instead of QDs and CdSe/ZnS monolayer, respectively."
We have taken the reviewer's comment into account and replaced QDs and CdSe/ZnS monolayer by CdSe/ZnS QDs and CdSe/ZnS QD monolayer in Table 1 and Table 2.
Reviewer 2 Report
In this manuscript, the authors found that a monolayer of CdSe QDs could double the photoresponse amplitude of multilayer graphene and doesn't influence its photoresponse characteristic time. Besides that, the authors also found that the average rate of energy/charge transfer from QDs to the MLG can be estimated via photoinduced processes on the QD surface. The research is interesting. I would like to recommend its publication after addressing the following questions.
- The detailed characterization (such as SEM and TEM) should be provided in the revised manuscript.
- How does the size of QDs affect the performance?
- How does the thickness (beside 30-40nm) affect the performance?
- New one is required for figure 4b.
- A lot of journals’ abbreviation are not right. Please correct them
- Some related references should be included in the revised manuscript: Adv. Mater. 2019, 31, 1802403; Angew Chem Int. Ed. 2015, 54, 8425.
Author Response
Comment 1
“The detailed characterization (such as SEM and TEM) should be provided in the revised manuscript.”
SEM image of MLG/QD structure has been added to SEM images of QD monolayer and MLG in Figure S2 in Supplementary Information file.
Comment 2
“How does the size of QDs affect the performance?"
It's well known that a decrease in QD size leads to an increase in electron transfer rate in QD-based hybrid structures due to increase in driving force (see for example J. Am. Chem. Soc. 2008, 130, 12, 4007-4015; https://doi.org/10.1021/ja0782706). At the same time, no direct correlation between the electron transfer rate and the performance of QD-based hybrid structures has been demonstrated so far because there are multiple parameters of the structures, such as the type and concentration of QD stabilizers, the trap states number on the QD surface, etc. that can significantly influence electron transfer efficiency. Therefore, the difference in these parameters of QDs with different sizes can totally eliminate the dependence of electron transfer efficiency on QD size in QD-based hybrid structures. In our work, we have used the QD ensemble with 5.5 nm average core size only and now we don't have any experimentally confirmed assumption concerning how QD size will influence MGL/QD structure performance.
Comment 3
“How does the thickness (beside 30-40nm) affect the performance?”
It's well known that the MLG thickness affects the degree of the graphene work function blurring, as well as the resistance of graphene and charge carrier mobility. Therefore, these parameters of MLG samples can affect the structure performance in opposite ways, and comparative study of the structures with different thickness of MLG is needed to reveal the dependence of the MLG/QD structure performance on MLG thickness.
Comment 4
“New one is required for figure 4b.”
We have revised Fig 4.b according to the reviewer's comment.
Comment 5
"A lot of journals’ abbreviation are not right. Please correct them"
We have checked and fixed the reference list.
Comment 6
"Some related references should be included in the revised manuscript: Adv. Mater. 2019, 31, 1802403; AngewChemInt. Ed. 2015, 54, 8425"
We have included these papers in the reference list as [24] and [25].